# Effects of Edible Organic Acid Soaking on Color, Protein Physicochemical, and Digestion Characteristics of Ready-to-Eat Shrimp upon Processing and Sterilization

**DOI:** 10.3390/foods13030388

**Published:** 2024-01-24

**Authors:** Chao Guo, Yingchen Fan, Zixuan Wu, Deyang Li, Yuxin Liu, Dayong Zhou

**Affiliations:** State Key Laboratory of Marine Food Processing & Safety Control, National Engineering Research Center of Seafood, Collaborative Innovation Center of Seafood Deep Processing, School of Food Science and Technology, Dalian Polytechnic University, Dalian 116034, China; gc823705522@163.com (C.G.); fanyingchen723@163.com (Y.F.); zixuanwu_0609@163.com (Z.W.); dpuldy@163.com (D.L.); forever--xin@126.com (Y.L.)

**Keywords:** ready-to-eat shrimp, processing and sterilization, color, organic acids, in vitro simulated digestion

## Abstract

Soft-packed ready-to-eat (RTE) shrimp has gradually become popular with consumers due to its portability and deliciousness. However, the browning caused by high-temperature sterilization is a non-negligible disadvantage affecting sensory quality. RTE shrimp is processed through “boiling + vacuum soft packing + high temperature and pressure sterilization”. Ultraviolet-visible (UV) spectroscopy with CIELAB color measurement showed that phytic acid (PA) + lactic acid (LA), PA + citric acid (CA), and PA + LA + CA soaking before cooking alleviated browning, as well as UVabsorbance and the browning index (BI). Meanwhile, UV spectroscopy and fluorescence spectroscopy showed that organic acid soaking reduced the content of carbonyl, dityrosine, disulfide bonds, surface hydrophobicity, and protein solubility, but promoted the content of free sulfhydryl and protein aggregation. However, in vitro digestion simulations showed that organic acid soaking unexpectedly inhibited the degree of hydrolysis and protein digestibility. This study provides the basis for the application of organic acids as color protectors for RTE aquatic muscle product.

## 1. Introduction

Pacific white shrimp (*Litopenaeus Vannamei*), belonging to *Arthropoda*, *Crustacea*, *Decapoda*, *Penaeidae*, *Penaeus*, is one of the three largest farmed shrimps in the world. As an aquatic muscle food, it is rich in protein, polyunsaturated fatty acids, vitamins, and minerals, and is popular with consumers [1].

At present, Pacific white shrimp is mainly sold fresh and frozen, supplemented by drying, etc. Selling the product fresh, as a traditional selling method, can fully retain the nutritional quality. However, due to the rich nutrients, high water content, and strong endogenous enzyme activity, it is prone to deterioration and blackening, making for geographical restrictions. Selling the product frozen is the most widely utilized method of sale, which can effectively prevent spoilage caused by microorganisms and enzymes. However, selling it frozen requires strict cold chain control throughout the process, which makes the cost high [2]. Dried shrimp is a form of product that has emerged in recent years due to its unique flavor, stable quality, and convenient consumption. However, due to the low moisture content, dried shrimp tastes poor [3]. Recently, soft-packed ready-to-eat (RTE) products have gradually emerged due to their ease of preparation and consumption. RTE shrimp is prepared with a “boiling + vacuum packing + thermal sterilization” process [4]. Compared to dried shrimp, RTE shrimp has a higher moisture content and good taste. The sensory properties of RTE shrimp, which include many factors such as color, flavor, and texture, are of great importance to consumers [5]. Due to the action of astaxanthin, boiled shrimp has a bright pink color. However, the Maillard reaction that occurs during thermal sterilization and storage darkens the shrimp color. Wang S. et al. [6] found that the color of roasted Pacific white prawn was deepened by thermal sterilization. Li et al. [4] found that RTE shrimp changed from orange–red to dark brown during storage.

Phytic acid (PA) is a kind of edible organic acid, which can inhibit oxidation reaction by complexing metal ions [7]. Lactic acid (LA) and citric acid (CA) are edible acidity regulators, which have the functions of maintaining food color, improving food flavor, and assisting antioxidant [8,9,10]. Our previous study found that PA combined with LA could effectively delay the color deterioration of RTE shrimp during storage [4]. However, the effect of organic acids on the color of RTE products upon high-temperature sterilization has rarely been studied.

In addition to delaying color deterioration, organic acids can also bind to protein, causing changes in protein physicochemical and digestive properties. PA is rich in phosphate ions, which can form binary complexes with amino groups of protein through electrostatic interactions at low pH and form ternary complexes with protein through cationic bridges in an environment above the isoelectric point [11]. CA contains a carboxyl group, in which the positively charged carbon can cross-link with the amino group of the protein to form an amide bond through a nucleophilic substitution reaction [12]. Kaspchak et al. [13] found that protein aggregation caused by PA binding to proteins through electrostatic interaction reduced the digestibility of bovine serum albumin in vitro. Qiu et al. [14] found that in the stomach environment (acidic pH), CA reduced digestibility by promoting the aggregation of wheat gluten protein, while in the intestinal environment (neutral pH), CA improved digestibility by significantly increasing the solubility of wheat gluten protein. However, few studies have addressed the effect of organic acids on protein physicochemical properties and the protein digestibility of RTE aquatic muscle products.

Therefore, the aim of this study was to investigate the effects of organic acid soaking on color, protein physicochemical properties, and the in vitro digestibility of RTE shrimp upon cooking and sterilization. This work will provide a theoretical basis for the color regulation, protein physicochemical properties, and digestive characteristics of RTE aquatic muscle products.

## 2. Materials and Methods

### 2.1. Materials and Chemicals

Frozen Litopenaeus vannamei shrimp (about 9 g per shrimp) was purchased from a market next to Dalian Polytechnic University in Dalian, Liaoning, China. PA, CA, and a phosphorus-free water-retaining agent (all food-grade additives) were purchased from Zhejiang Yinuo Biotechnology Co., Ltd. (Lanxi, China). LA was purchased from Hebei Wanbang Industrial Co., Ltd. (Xingtai, China). Other reagents in this study were purchased from Bonuo Reagent Co., Ltd. (Dalian, China).

### 2.2. Sample Preparation

After thawing (immersed in distilled water at about 15 °C) for 2 h, the shrimps were peeled and decapitated and then divided into 4 groups, including the control group (group 1) and three organic acid soaking groups (group 2–4). The shrimps in group 1 were soaked in distilled water (1:2, *w*/*v*) for 3 h at 4 °C and the shrimps in group 2–4 were soaked in 0.04% PA + 0.2% LA solution, 0.04% PA + 0.2% CA solution, and 0.04% PA + 0.2% LA + 0.2% CA solution (1:2, *w/v*) for 3 h at 4 °C, respectively. Subsequently, the RTE shrimp was processed by boiling in the corresponding soaking solution at 100 °C for 7 min, drying at 40 °C for 1 h, vacuum packing, and then sterilizing at 102.9 kPa, 115 °C for 20 min.

### 2.3. Sensory Evaluation and Color Analysis

The color evaluation (redness, yellowness, brightness, glossiness, and overall acceptability) of the RTE shrimp was conducted by a trained panel of 8 researchers. On a scale of 1–30 points, a sample grade was assigned, where 30 points represented strongly liking and 1 point represented strongly disliking the shrimp. The color of the RTE shrimp was determined with a HunterLab UltraScan PRO colorimeter (Hunter Associates Laboratory, Inc., Reston, VA, USA) combined with a three-point test [15].

### 2.4. Ultraviolet Absorbance (UV-Absorbance) and Browning Index (BI)

The UV-absorbance and BI were measured according to the method described by Ajandouz et al. [16].

### 2.5. Thiobarbituric Acid-Reactive Substances (TBARs)

TBARs were determined based on the method of Li et al. [17].

### 2.6. Protein Oxidation

#### 2.6.1. Carbonyl

The carbonyl was determined with a protein carbonyl assay kit (No. A087, Jiancheng Technology Co., Nanjing, China) by reacting protein carbonyl with 2,4-dinitrophenylhydrazine to form a red 2,4-dinitrophenylhydrazone precipitate. The results were expressed in nanomoles per milligram of protein.

#### 2.6.2. Free Sulfhydryl

The free sulfhydryl (SH) was determined based on the reaction of DTNP with sulfhydryls [18].

### 2.7. Protein Cross-Linking

#### 2.7.1. Dityrosine

Dityrosine was determined according to the method of Ma et al. [19]. The content of dityrosine in the protein solution was recorded with a hitachIF-2700 fluorescence spectrophotometer (Hitachi Co., Tokyo, Japan) (emission wavelength 420 nm, excitation wavelength 325 nm, and slit 5 nm).

#### 2.7.2. Disulfide Bond

The disulfide bond was determined with the method of Beveridge et al. [18] based on the reaction of DTNP with sulfhydryls.

### 2.8. Protein Aggregation

#### 2.8.1. Salt-Soluble Protein Aggregation

Protein aggregation was measured with the fluorescence intensity emitted by Nile red binding to protein aggregates [20]. An amount of 3 mL of the sample supernatant (the protein concentration was adjusted to 2 mg/mL) was mixed with 30 μL of Nile red (0.32 mg/1 mL of ethanol), vortexed for 1 min, and the fluorescence was measured with a hitachIF-2700 fluorescence spectrophotometer (Hitachi Co., Japan) (emission wavelength 620 nm, excitation wavelength 560 nm, and slit 10 nm). The results were expressed in arbitrary units (AU).

#### 2.8.2. Salt-Soluble Protein Solubility

Protein solubility was measured with the method of Vossen and De Smet [21].

### 2.9. Surface Hydrophobicity

Protein surface hydrophobicity was measured with 8-aniline-1-naphthalene sulfonic acid (ANS) [22]. The sample protein concentration was adjusted to 0.025, 0.05, 0.1, 0.15, and 0.2 mg/mL. Then 40 μL of 8 mM ANS was added into the 4 mL protein solution. After mixing, the fluorescence intensity was measured with a hitachIF-2700 fluorescence spectrophotometer (Hitachi Co., Japan) (emission wavelength 470 nm, excitation wavelength 390 nm, and slit width 5 nm). Surface hydrophobicity was obtained with the slope of a plot of fluorescence intensity versus protein concentration.

### 2.10. In Vitro Digestion

The shrimp freeze-dried powder was simulated for digestion in vitro [23]. An amount of 0.5 g of freeze-dried shrimp powder was digested with 3 mL of simulated gastric fluid (containing 2000 U/mL pepsin, pH 3.0) at 100 rpm at 37 °C for 2 h. Then 6 mL of simulated intestinal fluid (containing 100 U/mL trypsin, pH 7.0) was added to the system to digest at 100 rpm at 37 °C for 2 h. Digestion was terminated by boiling for 10 min.

#### 2.10.1. Degree of Hydrolysis (DH)

The DH was measured with ortho-phthalaldehyde (OPA) [24]. Briefly, digested samples were centrifuged at 10,000× *g* for 10 min. Then, 200 μL of the supernatant was mixed with 1.5 mL of OPA reagent for 2 min, and the absorbance was immediately measured at 340 nm. At the same time, the standard samples (serine) and blank samples were taken for measurement. The DH was calculated with the following formula:$${\mathrm{Seirne-NH}}_{2}(\mathrm{mmol}/L)=\frac{{\mathrm{OD}}_{\mathrm{sample}}-{\mathrm{OD}}_{\mathrm{blank}}}{{\mathrm{OD}}_{\mathrm{standard}}-{\mathrm{OD}}_{\mathrm{blank}}}\times 0.9516\times A\times B\times 0.001÷m÷P$$
where OD is the absorbance value, A is the volume of the supernatant, B is the diluted multiples, m is the sample mass, and P is the protein content in the sample (percent).
$$h=\frac{{\mathrm{Seirne-NH}}_{2}-\beta }{\alpha }\;\mathrm{meqv}/g\;\mathrm{protein}$$
where α is 1.00 and β is 0.40.
$$\mathrm{DH}\;(\%)=\frac{h}{h_{tot}}\;\times \;100$$
where h is the number of hydrolyzed peptide bonds and h*_tot_* is the total number of peptide bonds.

#### 2.10.2. Dry Matter Digestibility

Dry matter digestibility was measured according to the method of Fang et al. [25]. In short, the digested samples were mixed with ethanol at a ratio of 1:3 (*m*/*v*) to precipitate protein and centrifuged at 10,000× *g* for 10 min. The supernatant was discarded, and the precipitate was freeze-dried. The digestibility was calculated with the following formula:$$\mathrm{Dry}\;\mathrm{matter}\;\mathrm{digestibility}\;(\%)=\frac{P_{1}-P_{2}}{P_{1}}\times 100$$
where P_1_ is the dry matter weight before digestion and P_2_ is the dry matter weight of the freeze-dried precipitate after digestion.

### 2.11. Statistical Analysis

Three parallel experiments were carried out in the above experiments. A one-way analysis of variance (ANOVA) with SPSS software (version 26.0, SPSS Inc., Armonk, NY, USA) was used to assess the differences between the means. *p* < 0.05 was considered a statistically significant difference. Origin 2018 (OriginLab, Inc., Northampton, MA, USA) software was used for data mapping.

## 3. Results and Discussion

### 3.1. Sensory Evaluation and Color Analysis

The raw shrimp appeared gray–blue due to astaxanthin binding to protein [26]. After cooking, the structure of the astaxanthin protein was destroyed and the red astaxanthin was released, causing the shrimp to turn pink–red (Figure 1A) [27]. Meanwhile, sensory evaluation based on redness, yellowness, brightness, gloss, and overall acceptability showed that the color of the boiled shrimp was red and bright, with a higher overall acceptability (Figure 1G). However, after high-temperature sterilization, the shrimp visually turned brown and dark (Figure 1B), with a lower overall acceptability (Figure 1G). In contrast, the shrimp soaked in organic acids had a higher overall acceptability.

The color parameters of the RTE shrimp in different groups upon thermal processing are shown in Figure 1C–F. L* is the brightness index: the larger the value, the better the sample gloss. a* and b* are the color indexes, where a* represents red (+) or green (−) and b* represents yellow (+) or blue (−). The W* value is the whiteness value, which is proportional to the whiteness of the sample [28]. Compared with the control group, the PA + LA-, PA + CA-, and PA + LA + CA-soaked groups showed higher L* and W* but lower a* and b*, indicating that organic acid soaking inhibited the color deterioration of the shrimp upon sterilization. Qi et al. [15] also found that CA (0.1–0.5% *w*/*t*) treatment can significantly improve the overall brightness of seasoned fish after 35 min of sterilization at 110 °C.

### 3.2. UV-Absorbance and BI

The Maillard reaction is a non-enzymatic process in which melanoidins are formed from carbonyls (reduced sugar) and amino compounds (amino acid and protein) through a series of reactions [29]. The Maillard reaction greatly contributes to the black–brown color of shrimp, and its intermediate and final products can be characterized with UV-absorbance and BI, respectively. UV-absorbance (A294 nm) is an indicator to characterize colorless intermediate products produced by the Maillard reaction [30], while the BI (A420 nm) is an indicator of melanoid formation at the end of the Maillard reaction [31].

The A294 nm and A420 nm values of RTE shrimp in the different groups upon thermal processing are shown in Figure 2A,B. Compared to the control group, the A294 nm and A420 nm values were reduced in PA + LA-, PA + CA-, and PA + LA + CA-soaked groups, suggesting that organic acid soaking had a certain inhibitory effect on the Maillard reaction. The Maillard reaction rate is affected by the pH value of the system, which is positively correlated with the pH value in the range of pH 3~9 [32]. The possible reasons are as follows: (1) at low pH, the amino group is in a protonated state (-NH_3_^+^), which is not conducive to carbonyl-amide condensation between sugar and amino acids. While at high pH, the amino group is completely deprotonated, which accelerates the carbonyl ammonia condensation [33]; (2) at a low pH, reduced sugars tend to exist in a closed loop, which shields the carbonyl group to a certain extent and is not conducive to the Maillard reaction, while a high pH value provides favorable conditions for the molecular rearrangement of sugar, which can promote the occurrence of nucleophilic addition reaction [34]. Therefore, it was speculated that the low pH environment caused by PA + LA-, PA + CA-, and PA + LA + CA-soaking was one of the reasons for inhibiting the Maillard reaction. In addition, metal ions have been shown to accelerate the Maillard reaction by catalyzing the oxidation of lipids to aldehydes (substrates of the Maillard reaction) and by accelerating the formation of Amadori compounds (the product of the Maillard reaction primary stage) [35]. In this study, compared with the control group, organic acid soaking inhibited the oxidation of lipids to aldehydes (TBARs, Figure 2C), resulting in a decrease in the substrate of the Maillard reaction. PA contains six phosphate groups, which are highly electronegative and can chelate metal ions by ionic bonding, while CA and LA contain three and two carboxyl groups, respectively, which can form coordination bonds with metal ions to form stable complexes [36,37]. Sanchis et al. [38] found that PA can inhibit the formation of advanced glycosylation end products (AGEs) in Fe^3+^, Lys, Arg, and ribose systems by binding to iron ions. Therefore, it is speculated that the formation of complexes between organic acids and metal ions is another reason to inhibit the Maillard reaction.

### 3.3. Protein Oxidation

Protein oxidation deepens the color of aquatic muscle food. In addition, protein oxidation also has an effect on protein digestibility. Protein carbonylation is a typical feature of protein oxidation. Carbonyl groups can be formed by direct oxidation of lysine, arginine, proline, and threonine residues on the protein side chain by free radicals. In addition, carbonyl groups can also be formed at the fracture site by the direct action of free radicals on peptide bonds by extracting hydrogen from carbon atoms [39]. The reduction in free sulfhydryl groups is another important feature of protein oxidation. The sulfur outer layer on the sulfhydryl group has many lone pairs of electrons, which exhibit strong nucleophilicity and are easily attacked by free radicals to generate oxidation products such as disulfide bonds [40].

The oxidation indexes of RTE shrimp from different groups upon thermal processing are shown in Figure 3A,B. Compared with those of the control group, the carbonyl contents of the PA + LA-, PA + CA-, and PA + LA + CA-soaked groups decreased, while the free sulfhydryl increased, indicating that organic acid soaking could inhibit protein oxidation. Studies have shown that thermal processing promotes the oxidation of protein in meat products, which is manifested as the increase in carbonyl groups and the decrease in free sulfhydryl groups [41]. The reasons for protein oxidation caused by thermal processing are as follows: (1) thermal processing reduces the antioxidant activity of antioxidant components (such as catalase, glutathione peroxidase, etc.) [42]; (2) thermal processing destroys the cell membrane structure, and transition metal ions such as Cu^2+^ and Fe^2+^ are released by cells, inducing free radical production [43]. Therefore, it was speculated that PA + LA, PA + CA, and PA + LA + CA soaking could interrupt or destroy the chain reaction of protein oxidation by chelating metal ions.

### 3.4. Protein Cross-Linking

Protein cross-linking refers to the process of forming cross-links between polypeptide chains (intramolecular or intermolecular) through covalent bonds, which affect the protein digestive properties by affecting the contact site with digestive enzymes [44]. Dityrosine and disulfide bonds are covalent bonds formed by the oxidation of tyrosine and cysteine within or between molecules, which can be used to characterize protein cross-linking [45]. Tyrosine is an amino acid that is easily attacked by free radicals. As a sensitive site in tyrosine, the phenolic hydroxyl group is attacked by free radicals to form phenoxy radicals. Phenoxy radicals are isomerized to C-centric radicals, which are structurally unstable and eventually exist as stable dimerizers (dityrosine) [46]. Fluorescence spectroscopy is the most commonly used method for detecting dityrosine. The singly ionized form of dityrosine has strong fluorescence at about 400 nm after excitation at about 315 nm, and its fluorescence intensity is proportional to the content of the dityrosine [47]. Cysteine is another amino acid that is sensitive to free radicals. The reactive sulfhydryl group on the cysteine side chain residue is easily attacked by free radicals and easily forms covalent bonds with the other half of cysteine side chain residues [48].

The dityrosine and disulfide bonds of RTE shrimp upon thermal processing are shown in Figure 4A,B. Compared with those of the control group, the dityrosine and cysteine contents of the PA + LA-, PA + CA-, and PA + LA + CA-soaked groups were significantly reduced, indicating that organic acid soaking had a certain inhibitory effect on protein covalent cross-linking. Previous studies have shown that tyrosine and cysteine are easily attacked by free radicals to form dityrosine and disulfide bonds upon thermal processing [19]. In this study, it was hypothesized that the metal ion chelation ability of organic acids would weaken the oxidation of dityrosine and cysteine, thereby inhibiting the covalent cross-linking of proteins. 

### 3.5. Salt-Soluble Protein Aggregation and Salt-Soluble Protein Solubility

As an aquatic muscle food, shrimp protein is composed of salt-soluble protein (myofibrillar protein), water-soluble protein (sarcoplasmic protein), and muscle-matrix protein [49]. The main component of salt-soluble protein is myofibrillar protein, which is the main protein in shrimp muscle, accounting for 65% to 75% of the total protein [50]. Aggregation and solubility affect protein digestibility by affecting the physical accessibility of proteins and digestive enzymes.

The fluorescence intensity of salt-soluble protein is shown in Figure 5A. Compared with the control group, the PA + LA-, PA + CA-, and PA + LA + CA-soaked groups all showed significantly enhanced fluorescence intensity, implying that organic acid soaking promoted the aggregation of salt-soluble proteins. Studies have shown that forces that drive protein aggregation include covalent bonds caused by protein oxidation (dityrosine and disulfide bonds) [48], covalent bonds formed by proteins to other non-protein substances (amide, etc.) [51], and weak interactions (electrostatic interactions, van der Waals forces, hydrophobic interaction, etc.) [52]. The above results showed that PA + LA, PA + CA, and PA + LA + CA soaking can inhibit the formation of dityrosine and disulfide bonds (covalent bonds caused by protein oxidation) by chelating metal ions. Therefore, it is reasonable to speculate that covalent bonds formed by cross-linking protein with other non-protein substances and the weak interaction may have been responsible for the increased protein aggregation in the PA + LA-, PA + CA-, and PA + LA + CA-soaked groups. CA contains three carboxyl groups, which can form covalent bonds through amide bonds with the the free amino group of the protein by nucleophilic substitution to produce a protein–CA conjugate [53]. PA contains abundant phosphate ions, which can form binary complexes with amino groups of protein through electrostatic interaction at low pH and form ternary complexes with proteins through cationic bridges in an environment above the isoelectric point [11]. Darby et al. [54] also found that phytate–protein lysozyme binding interactions were driven by electrostatic interactions between positively charged proteins and negatively charged phytates. In addition, PA, LA, and CA, as edible organic acids, reduce the pH of the system by dissociating H^+^. Compared with that in the control group, the pH values in the systems of the PA + LA-, PA + CA-, and PA + LA + CA-soaked groups were closer to the isoelectric point of myofibrillar protein (the main protein in shrimp), making the protein less charged on the surface and more likely to aggregate.

Protein solubility refers to the amount of protein remaining in the supernatant after centrifugation in a specific solution [55]. The solubility of protein is inversely proportional to the degree of aggregation. The solubility of the salt-soluble protein of RTE shrimp upon thermal processing is shown in Figure 5B. Compared with the control group, the PA + LA-, PA + CA-, and PA + LA + CA-soaked groups all showed decreased solubility of salt-soluble protein. Previous studies have also shown that organic acids can reduce the solubility of protein. Kaspchak et al. [56] treated soybean, pea, and rice protein isolates with PA and found that PA reduced the solubility of the three isolates at pH 3.0. Li et al. [57] treated whey protein isolate with CA and found that CA significantly reduced the solubility of the whey protein isolate.

### 3.6. Surface Hydrophobicity

Surface hydrophobicity indicates the exposure of hydrophobic amino acid residues in protein molecules and is an index to characterize the change of protein structure. A common reagent is 40 ANS for detecting surface hydrophobicity, which is basically non-fluorescent in aqueous solutions and enhanced after highly specific binding to hydrophobic groups [58].

The surface hydrophobicity of RTE shrimp upon thermal processing is shown in Figure 5C. Compared with the control group, the PA + LA-, PA + CA-, and PA + LA + CA-soaked groups all significantly showed reduced surface hydrophobicity. The oxidation of protein induced by thermal processing exposed the internal hydrophobic amino acids, which led to an increase in surface hydrophobicity [59]. In this study, it was speculated that PA + LA, PA + CA, and PA + LA + CA soaking can inhibit protein oxidation with chelating metal ions, thereby reducing the surface hydrophobicity of the protein. In addition, studies have shown that protein aggregation may shield some hydrophobic amino acids, which leads to a decrease in surface hydrophobicity [57]. Our results showed that PA + LA, PA + CA, and PA + LA + CA soaking promoted protein aggregation, which was speculated to be another reason for the decrease in surface hydrophobicity.

### 3.7. In Vitro Digestion

The DH refers to the degree to which a protein is hydrolyzed into peptides and amino acids under the action of digestive enzymes [60]. In this study, the main substances digested in the system were proteins due to the use of pepsin and trypsin for in vitro simulated digestion. Dry matter digestibility was used to reflect the degree of protein digestion. The DH and dry matter digestibility of RTE shrimp upon thermal processing are shown in Figure 6A–D. Compared with that in the control group, organic acid soaking significantly decreased the simulated gastric and gastrointestinal DH and dry matter digestibility.

The difference in protein digestibility between the control group and PA + LA-, PA + CA-, and PA + LA + CA-soaked groups was related to the difference in protein structure. Lysine and arginine residues are the most common oxidation sites of proteins and are also the specific cleavage sites of trypsin [61]. This indicates that the degree of protein oxidation is inversely proportional to digestibility to some extent. However, in this study, through a correlation analysis (Figure 6E), it was found that the DH and digestibility of RTE shrimp were positively correlated with protein oxidation indices and negatively correlated with protein aggregation. Therefore, it was speculated that the protein aggregation induced by PA + LA, PA + CA, and PA + LA + CA soaking led to the decrease in DH and digestibility. The phosphoric acid of PA and the carboxyl group of LA and CA could react with proteins through electrostatic interactions and covalent bonds. This caused protein aggregation and blocked the protein–enzyme binding site, resulting in reduced protein digestibility. Kaspchak et al. [13] found that PA reduced the in vitro digestibility of bovine serum albumin, possibly due to protein aggregation caused by electrostatic interactions. Qiu et al. [14] explored the effect of CA on the physicochemical properties and in vitro digestibility of wheat gluten and found that in the stomach environment (acidic pH), CA decreased digestibility by promoting the aggregation of wheat gluten protein, while in the intestinal environment (neutral pH), CA increased digestibility by significantly improving the solubility of wheat gluten protein.

## 4. Conclusions

The browning of RTE shrimp upon sterilization could be alleviated by PA + LA, PA + CA, and PA + LA + CA soaking. Meanwhile, organic acid soaking inhibited protein oxidation and protein cross-linking by chelating metal ions, but induced protein aggregation by binding to proteins with covalent bonds and weak interactions. However, it was found that organic acid soaking had a negative effect on protein digestion in vitro, which may have been attributed to the increase in aggregation. This study provides the basis for the application of organic acids as color protectors for RTE aquatic muscle products. In the future, regulatory methods that can protect the color of RTE aquatic muscle products without affecting protein digestibility, such as adding lactic acid bacteria instead of organic acids, should be explored.

## Figures and Tables

**Figure 1 foods-13-00388-f001:**
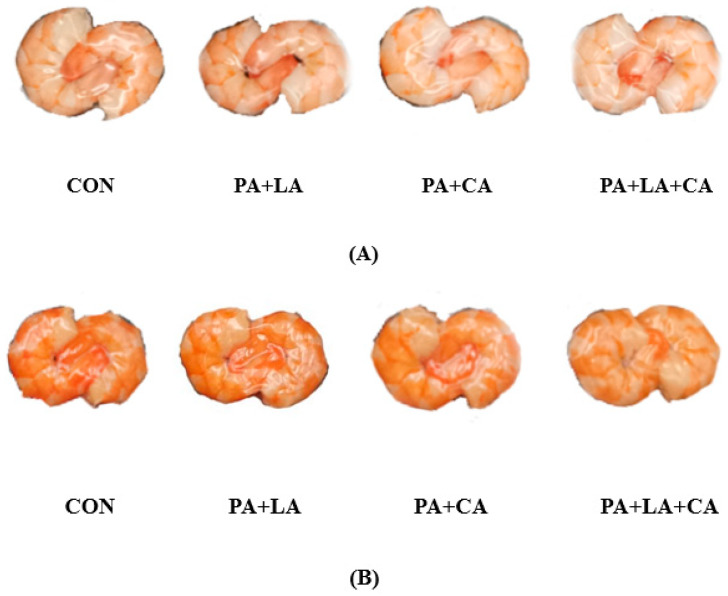
Pictures after boiling (**A**); pictures after sterilization (**B**); and changes in L* (**C**), a* (**D**), b* (**E**), W* (**F**), and sensory evaluation (**G**) of RTE shrimp with different treatments. Con, PA + LA, PA + CA, and PA + LA + CA represent shrimp soaked in different solutions (water, 0.04% PA + 0.2% LA solution, 0.04% PA + 0.2% CA solution, and 0.04% PA + 0.2% LA + 0.2% CA solution, respectively) at 4 °C for 3 h. Different letters indicate significant differences at *p* < 0.05.

**Figure 2 foods-13-00388-f002:**
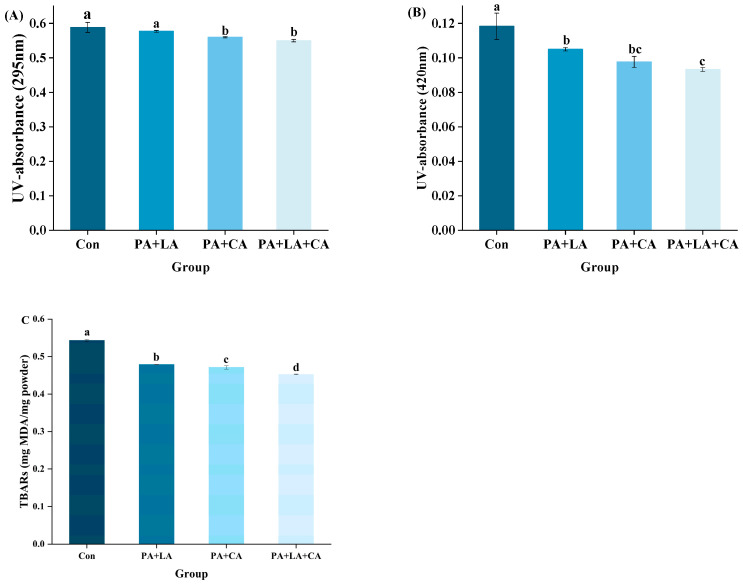
Changes in ultraviolet absorption A295 (**A**),browning index A420 (**B**), and TBARs (**C**) of RTE shrimp with different treatments. Con, PA + LA, PA + CA, and PA + LA + CA represent shrimp soaked in different solutions (water, 0.04% PA + 0.2% LA solution, 0.04% PA + 0.2% CA solution, and 0.04% PA + 0.2% LA + 0.2% CA solution, respectively) at 4 °C for 3 h. Different letters indicate significant differences at *p* < 0.05.

**Figure 3 foods-13-00388-f003:**
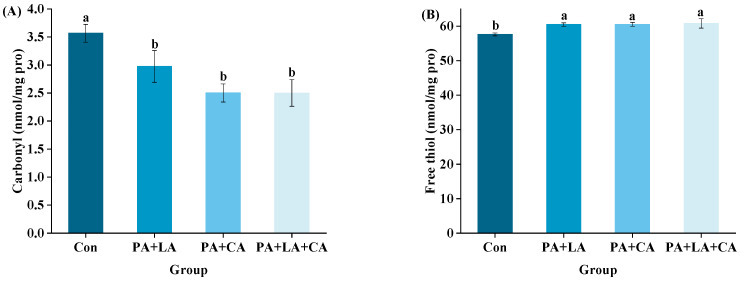
Changes in carbonyl content (**A**) and free thiol (**B**) of RTE shrimp with different treatments. Con, PA + LA, PA + CA, and PA + LA + CA represent shrimp soaked in different solutions (water, 0.04% PA + 0.2% LA solution, 0.04% PA + 0.2% CA solution, and 0.04% PA + 0.2% LA + 0.2% CA solution, respectively) at 4 °C for 3 h. Different letters indicate significant differences at *p* < 0.05.

**Figure 4 foods-13-00388-f004:**
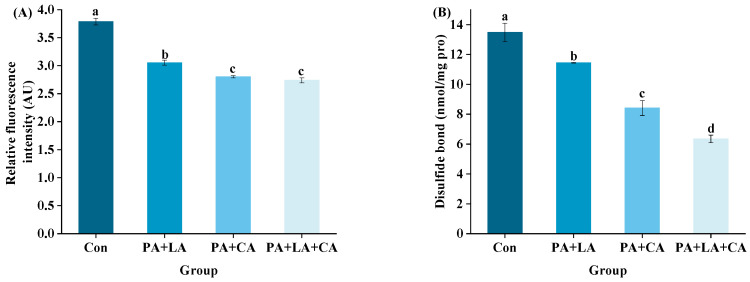
Changes in dityrosine (**A**) and disulfide bonds (**B**) of RTE shrimp with different treatments. Con, PA + LA, PA + CA and PA + LA + CA represent shrimp soaked in different solutions (water, 0.04% PA + 0.2% LA solution, 0.04% PA + 0.2% CA solution, and 0.04% PA + 0.2% LA + 0.2% CA solution, respectively) at 4 °C for 3 h. Different letters indicate significant differences at *p* < 0.05.

**Figure 5 foods-13-00388-f005:**
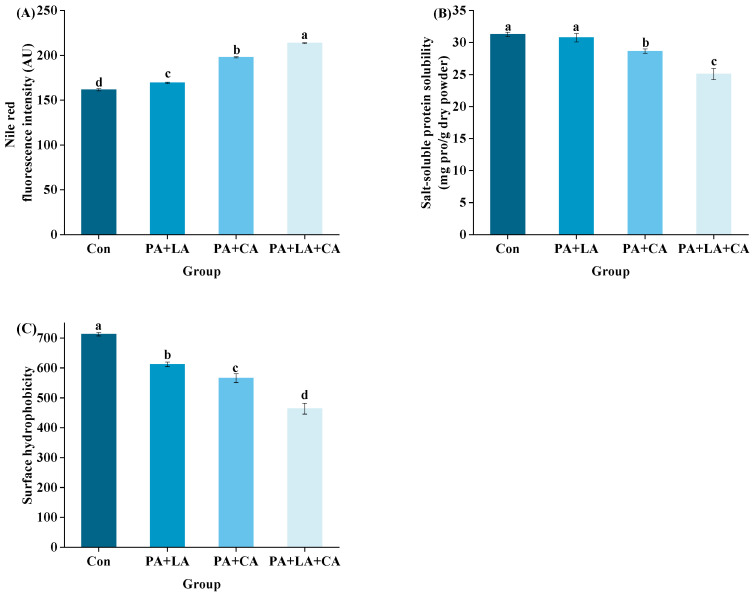
Changes in protein aggregation (**A**), salt-soluble protein solubility (**B**), and surface hydrophobic (**C**) of RTE shrimp with different treatments. Con, PA + LA, PA + CA, and PA + LA + CA represent shrimp soaked in different solutions (water, 0.04% PA + 0.2% LA solution, 0.04% PA + 0.2% CA solution, and 0.04% PA + 0.2% LA + 0.2% CA solution, respectively) at 4 °C for 3 h. Different letters indicate significant differences at *p* < 0.05.

**Figure 6 foods-13-00388-f006:**
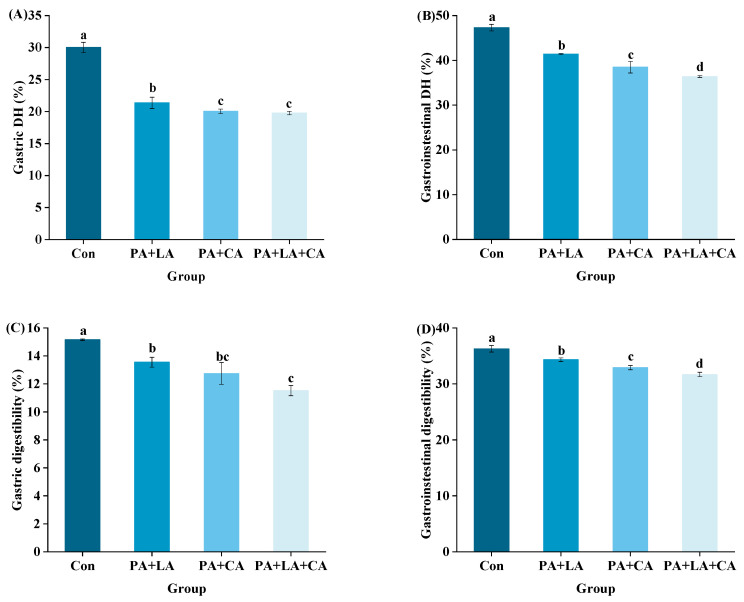
Changes in gastric DH (**A**), gastrointestinal DH (**B**), gastric digestibility (**C**), gastrointestinal digestibility (**D**), and correlation analysis (**E**) of RTE shrimp with different treatments. Con, PA + LA, PA + CA, and PA + LA + CA represent shrimp soaked in different solutions (water, 0.04% PA + 0.2% LA solution, 0.04% PA + 0.2% CA solution, and 0.04% PA + 0.2% LA + 0.2% CA solution, respectively) at 4 °C for 3 h. Different letters indicate significant differences at *p* < 0.05.

## Data Availability

Data are contained within the article.

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
