# Peer review of "Effects of Edible Organic Acid Soaking on Color, Protein Physicochemical, and Digestion Characteristics of Ready-to-Eat Shrimp upon Processing and Sterilization"

_foods, 2024, doi:10.3390/foods13030388_

Round 1

Reviewer 1 Report

Comments and Suggestions for Authors

The paper introduces a new method for prevention of browning or ready to eat shrimps. The paper is interesting and well presented. the main concern that was not investigatet is the influence of the treatment to the sensorz characteristics of shrimps. This aspect should be taken into consideration and explained in the text.

Reviewer 2 Report

Comments and Suggestions for Authors

Foods-2806527 Effects of Edible Organic Acids Soaking on Color, Protein physi-Cochemical and Digestion Characteristics of Ready-to-Eat Shrimps upon Processing and Sterilization

      This manuscript contains a significant amount of scientific data that can help protect processed shrimps from discoloration. An interesting method of treating fresh shrimps by soaking them in three different solutions of organic acids is applied. This was followed by cooking at the temperature of 100°C and drying at temperature of 40°C, vacuum packaging and sterilization at overpressure of 102.9 kPa, temperature of 115°C held for 20 minutes.

        The thus treated samples were subjected to color measurement, UV absorbance and browning index determination. Thiobarbituric acid reactive substances and protein oxidation products were determined. Furthermore, the authors dealt with protein crosslinking, their aggregation and solubility, surface hydrophobicity and in-vitro digestion. Part of this analysis was the determination of the degree of hydrolysis of DH and digestibility of dry matter.

       This extensive and carefully conducted experiment always compared treated samples with untreated ones, so it was possible to correctly state in the conclusion of the work that the study will provide a basis for the application of organic acids as color protection for ready to eat shrimps.

            I have only comments on the presented work listed below. After respecting them and editing the manuscript, I recommend the work for publication.

General comment

            I suggest to the authors whether in the next work they should withdraw from the use of organic acids as chemical compounds and use the method of protection of shrimps by the acidification process with the use of lactic bacteria and the addition of carbohydrates. These bacteria can grow very quickly and generate lactic acid, thus protecting the shrimps from the growth of pathogenic microorganisms. The reduced pH prevents the germination of spores and improves the taste for consumers without the need for heat treatment.

Specific comments

-          Line 3 word “physi-Cochemical” is probably typing mistake. I propose to use “physico-chemical” or “physicochemical”.

-          Line 49 you often use the name of the first author et al for in-text citations. but the citation [nr.] only at the end of the sentence. This is not in accordance with the rules. A parenthesis with the citation number should be placed immediately after "et al.” Move the bracket with the number 10 from line 51 just after "Kaspchak et al."

-          Line 51 there is Qiu et al. and its number is at line 55. Move it just behind “et al.”

-          Lines 143-149 digestibility of dry matter is tested by ethanol? I think there is need brief explanation.

-          Lines 169-172 contains reference Qi et al. But number 12 is at the end of line 172. Please do the shift behind “et al.”

-          Line 208 contains number “4”! Why?

-          Lines 210-212 contain reference Sanchis et al. with number 35 at the line 212. Please do the shift behind “et al.”

-          Line 287 contains number 45. If there is need to refer paper 45, there is need to add brackets [45].

-          Line 299 contains number 8.

-          Lines 299-302 contain the reference to Darby et al.. Number 51 is placed on line 302. Please shift the number after "et al."

-          Lines 313-314 contain reference Kaspchak et al.; please shift number 53 after „et al.“

-          Lines 314-316 contain reference Li et al.; please shift number 51 after „et al.“

-          Sections 3.5 and 3.6 comment on parts of Fig. 5 a, b and c. Figs. 5 d is not commented on here or anywhere else in the text. Please repair.

-          Lines 361-362 contain reference Kaspchak et al.; please shift number 10 after „et al.“

-          Lines 362-363 contain reference Qiu et al. There is missing number [11].

-          References: there are not given doi codes. I think it is compulsory to input them now.

Reviewer 3 Report

Comments and Suggestions for Authors

The authors investigated the effects of edible organic acids soaking on color, protein physiochemical and digestion characteristics of ready-to-eat shrimps upon processing and sterilization.It appears well written, and with useful results, and can be further improved by the following:

a) The introduction is somewhat scanty, and needs more background information, like 1) Storage quality attribute changes, and implications associated with shrimp products shortly after harvest (discuss them especially in the context of the parameters you have studied, and this should have a full paragraph of its own.
Also, what is the importance of ready to eat products, examples of them in the context of shrimp. All these should come before talking about phytic acid etc
b) Methods is ok
c) Results and discussion is also very ok. Authors are encouraged to connect the various attributes. How does color connect with the results of  uv absorbance, protein oxidation, etc etc, and vice versa. Try to make those connections, where literature avails.
d) Conclusions, please provide the direction for future studies ok
Look forward to your revised manuscript
